# Metal–Metal Correlation of Biodegraded Crude Oil and Associated Economic Crops from the Eastern Dahomey Basin, Nigeria

Saeed Mohammed [1] , Mimonitu Opuwari [1,*], Salam Titinchi [2,*] and Lilburne Cyster [3]

1    Department of Earth Sciences, Faculty of Natural Sciences, University of the Western Cape, Bellville 7535, Cape Town, South Africa; 3776487@myuwc.ac.za
2    Department of Chemistry, Faculty of Natural Sciences, University of the Western Cape, Bellville 7535, Cape Town, South Africa
3    Department of Biodiversity and Conservation Biology, University of the Western Cape, Bellville 7535, Cape Town, South Africa; lcyster@uwc.ac.za
*    Correspondence: mopuwari@uwc.ac.za (M.O.); stitinchi@uwc.ac.za (S.T.)

**Abstract:** The presence of heavy metals in plants from oil sand deposits may reflect mineralization resulting from petroleum biodegradation. Petroleum composition and heavy metal analyses were performed using thermal desorption gas chromatography and atomic absorption spectrophotometry on oil sand and plant root samples from the same localities in the Dahomey Basin. The results from the oil sand showed mainly heavy-end hydrocarbon components, humps of unresolved complex mixtures (UCM), absences of C6-C12 hydrocarbon chains, pristane, and phytane, indicating severe biodegradation. In addition, they showed varying concentrations of vanadium (2.699–7.708 ppm), nickel (4.005–11.716 ppm), chromium (1.686–5.733 ppm), cobalt (0.953–3.223 ppm), lead (0.649–0.978 ppm), and cadmium (0.188–0.461 ppm). Furthermore, these heavy metals were present in *Citrus*, *Theobroma Cacao*, *Elaeis guineensis*, and *Cola*. The chromium, nickel, vanadium, lead, cobalt, and cadmium concentrations in the *Citrus* were 7.475, 4.981, 0.551, 0.001, 0.806, and 0.177 ppm, respectively. For the *Theobroma Cacao*, the concentrations of chromium, nickel, vanadium, lead, cobalt, and cadmium were 7.095, 16.697, 2.151, 0.023, 3.942, and 0.254 ppm. *Elaeis guineensis* also showed the presence of chromium (32.685 ppm), nickel (32.423 ppm), vanadium (11.983 ppm), lead (0.190 ppm), cobalt (4.425 ppm), and cadmium (0.262 ppm). The amounts of chromium, nickel, vanadium, lead, cobalt, and cadmium in the *Cola* were 9.687, 9.157, 0.779, 0.037, 0.695, and 0.023 ppm. The World Health Organization's (WHO) safe and permissible limits for Cd (0.003 ppm), Cr (0.1 ppm), Ni (0.05 ppm), and Pb (0.1 ppm) in agricultural soils were all exceeded in the oil sand. The presence of these metals in the oil sands and their uptake by the plants could potentially be toxic, resulting in high mortality. The metal–metal correlation of the plant's rootsto the oil sand indicates the nonanthropogenic origin of the heavy metals, which leads to the conclusion that their source is related to the hydrocarbon accumulation in the Afowo sand.

**Keywords:** oil sand; eastern dahomey basin; economic crops; afowo formation; biodegradation; heavy metals

## 1. Introduction

The Cretaceous Composite [1] of the Eastern Dahomey Basin petroleum system is marked with oil sand deposits in the Cretaceous Afowo Formation of the Abeokuta Group. The Dahomey Basin is one of the basins in the Gulf of Guinea that evolved during rifting in the Late Jurassic to Early Cretaceous times [2,3]. It extends from southeastern Ghana through Togo and the Republic of Benin to southwestern Nigeria [4–6]. The eastern segment of the basin, also referred to as Eastern Dahomey Basin or Dahomey Embayment, contains Cretaceous to Recent sediments of up to 3000 m [7,8].

The oil sand deposits straddle four states (i.e., Lagos, Ogun, Ondo, and Edo) in southwestern Nigeria, with a reserve estimate of approximately 58 billion oil equivalent [9]. While the oil sand deposits have remained undeveloped [8], the region has fertile land, where economic crops, such as Theobroma cacao (cocoa), Cola (kola nut), Coffea arabica (coffee), Elaeis guineensis (oil palm), and Citrus (orange) thrive. Agriculture wasthe mainstay of Nigeria's economy, and the region was a major exporter of cocoa to the world before the discovery of petroleum in the Niger Delta Basin. Some of these economic crops are found within the areas of oil sand deposits.

Hydrocarbon generation results from the thermal break down of kerogen in the source rock. The kerogen's ultimate composition is influenced by the source materials (organic and inorganic) incorporated in the sediments. Some of these materials include heavy metals, which are released and transported during the generation and migration of the hydrocarbons from the source rock. The thermal evolution of Type I and II kerogen yields light crude oil with an API gravity of 30° to 40° [10]. Once the oil has accumulated in the reservoir, its quality can be drastically changed, and this degradation is attained through several processes and, in some instances, it can be so severe as to alter the characteristics of the crude oil forever. Degradation of the reservoired hydrocarbons results in the accumulation of the heavy-end molecular components with low API gravity, high viscosity, high sulfur, and high metal concentrations [11]. The close relationship in the concentration of heavy metals in crude oils reflect a genetic continuum in unaltered to completely biodegraded oils. Several metals have been reported in heavy oils; most of these metals are present only in trace amounts, and they bind with porphyrins to form organometallic complexes [12].

The types of metals found in oil sands are dependent on the geological environment from which the oil is formed. The migration of crude oil into porous and permeable sandstones where it is subsequently altered to heavy oil through the removal of light hydrocarbon fractions, results in the formation of oil sand. The initial crude oil is changed and degraded by processes such as biodegradation [13]. Some of the essential factors in the formation of oil sands include thermal maturation, biodegradation, and evaporative transformation [14]. Vanadium is one of the most abundant metals in crude oil, and its presence with other heavy metals has been reported to cause health problems [15]. In addition to crude oil alteration in the reservoir resulting in enriched metal content, its quality can be degraded during migration from the source rock [16].

Common metals found in the oil sand deposits are antimony, uranium, aluminium, tin, barium, gallium, silver, magnesium, sodium, molybdenum, zinc, cadmium, titanium, manganese, chromium, cobalt, arsenic, copper, lead, and iron [13]. Threshold concentrations for some heavy metals have been reported, and concentrations above these thresholds are known to cause health problems [17]. Heavy metal pollution constitutes one of today's most significant environmental problems [18]. The toxic pollutants released in waters can be harmful to both plants and animals. In addition, plants can absorb heavy metals from soils through their roots and leaves, causing morphological and physiological damage. Most heavy metals are toxic to humans, even at low concentrations. Examples of heavy metals that are highly toxic to plants and animals include cadmium, chromium, mercury, and lead [19]. It is, therefore, essential to identify, measure, and evaluate the concentrations and impact of heavy metal pollutants in these environments.

The present study aimed to evaluate the hydrocarbon composition of the oil sand, determine the extent of petroleum alteration, and establish and correlate heavy metal occurrences in the oil sand and some economic crops in the basin. The oil sands were analyzed for their hydrocarbon composition and heavy metals (i.e., Cr, Ni, V, Pb, Co, and Cd). In addition, *Citrus, Theobroma Cacao, Elaeis guineensis,* and *Cola,* were collected, prepared, and analyzed for chromium (Cr), nickel (Ni), vanadium (V), lead (Pb), cobalt (Co), and cadmium (Cd).

## 2. Materials and Methods

The present study focused on borehole samples of oil sands from the Dahomey Basin and the economic crops cultivated in the vicinity of the boreholes (Figure 1).

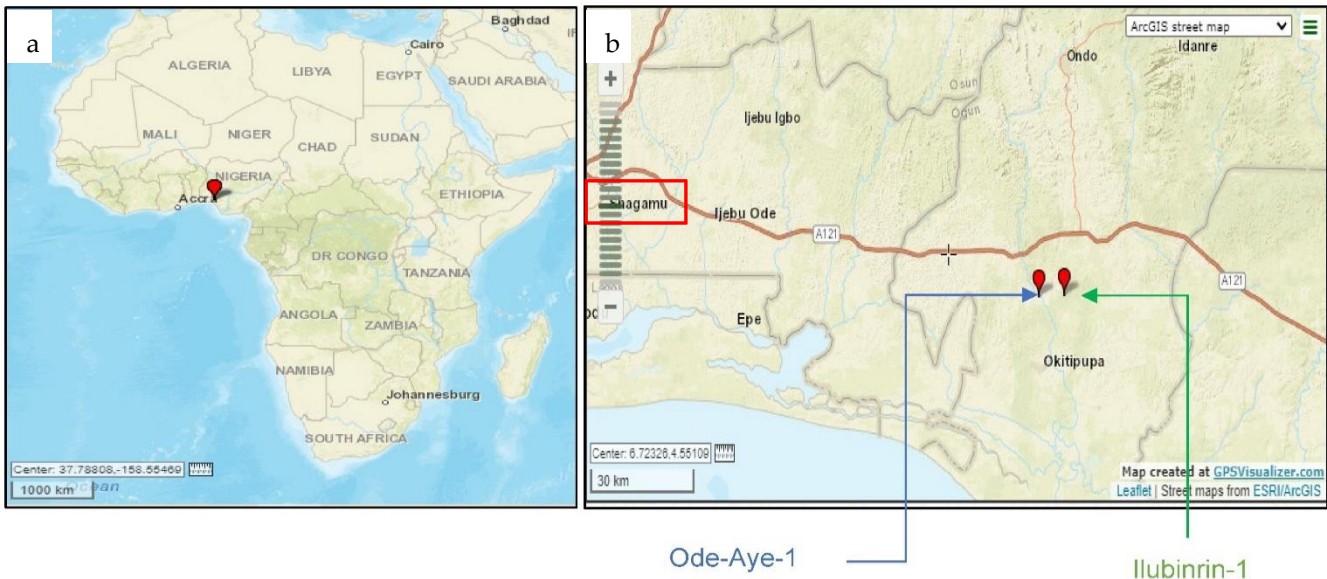

**Figure 1.** Maps showing the area of study. (**a**) Map of Africa showing the field of study in Nigeria. (**b**) Regional map of a part of southwestern Nigeria, indicating the well locations. The Ode-Aye-1 well is located at latitude 06°38′10.2″ N and longitude 004°45′59.7″ E. In contrast, Ilubinrin-1 is located at latitude 06°38′19.8″ N and longitude 004°49′47.9″ N. The maps were generated using G.P.S. Visualizer (G.P.S. Visualizer).

The Dahomey Basin is one of the sedimentary basins situated in the Gulf of Guinea. It extends from the eastern border of Liberia to the west edge of the Niger Delta Basin in Nigeria. The basins (marginal basins) in the province consist of the Ivory Coast, Tano, Saltpond and Central, Keta, and Benin Basins (Figure 2), and they have common structural and stratigraphic characteristics [1]; they are wrench-modified and host rocks ranging from the Ordovician to the Holocene.

The eastern boundary of the Gulf of Guinea Province is the Niger Delta Province [20] in southern Nigeria, and the western border is the West African Coastal Province.

The Gulf of Guinea evolved during the Late Jurassic to Early Cretaceous tectonism that was characterized by both block and transform faulting superimposed across an extensive Paleozoic basin during the breakup of the African, North American, and South American paleocontinents [1]. The tectonism and faulting initiated the separation of the thick continental crust of the African and South American plates forming divergent basins or pull-apart grabens separated by transform faults in the Early Albian time. Clastic, marine and, possibly, lacustrine sediments were deposited in the basins, forming the essential elements of a petroleum system. The province has undergone a complicated development. It is structurally defined by three major transform fault zones (Figure 2): the St. Paul Fracture Zone, the Romanche Fracture Zone, and the Chain Fracture.

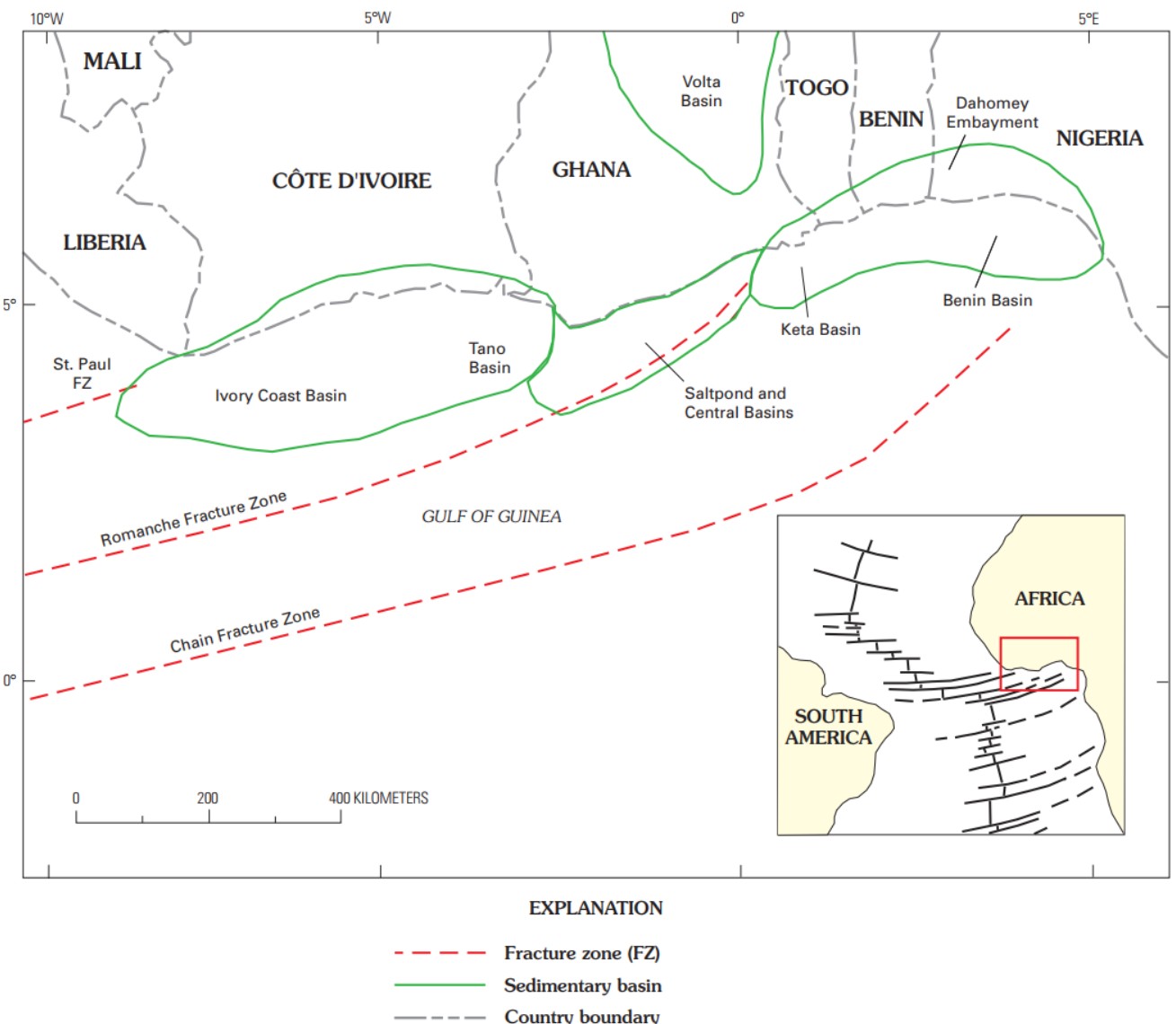

**Figure 2.** Sedimentary basins in the Gulf of Guinea Province of West Africa from the United States Geological Survey (https://pubs.usgs.gov/bul/2207/C/) (accessed on 2 May 2022) showing major structural features. The Mid-Atlantic Ridge and fractures are shown in the index map.

The stratigraphy of the Eastern Dahomey Basin (Table 1) has been discussed previously [21], and different classification schemes have been proposed [22]. However, regardless of the classification nomenclature, different stratigraphic names have been assigned for the same formation in other localities, leading to an obvious confusion [23].

As defined in [3], the stratigraphic successions of the Eastern Dahomey Basin comprise the Abeokuta Group and the Tertiary Formations of Ewekoro Akinbo, Oshosun, Ilaro, and Benin Coastal Plain Sands.

The Abeokuta Group comprises Ise, Afowo, and Araromi Formations. Ise Formation is the oldest formation in the Abeokuta Group. The Ise Formation consists of basal conglomerates, medium-to-coarse-grained sandstones with admixtures of kaolinitic clays [8]. The formation was assigned Valanginian–Barremian [21] and rests unconformably on the basement complex. Succeeding the Ise Formation is the petroliferous Afowo Formation. The Afowo Formation is hydrocarbon bearing, and it is composed of fine to medium and coarse-grained sandstone, shales, clays, and intercalations. The sandy facies are tar bearing, while the shales and clays are organic rich.

Table 1. Stratigraphy of the Eastern Dahomey Basin [23].

| Era | Period | Epoch | Stratigraphy of the Eastern Dahomey Basin | | |
|---|---|---|---|---|---|
| **Cenozoic** | Quaternary | Holocene Pleistocene | Jones and Hockey (1964) | Omatsola & Adegoke (1981) | |
| | | Oligocene | Coastal Plain Sands | Coastal Plain Sands | |
| | Tertiary | Eocene | Ilaro Formation | Ilaro Formation Ososhun Formation | |
| | | Paleocene | Ewekoro Formation | Akinbo Formation Ewekoro Formation | |
| **Mesozoic** | Cretaceous | Late to Early | Abeokuta Formation | Abeokuta Group | Araromi Formation Afowo Formation Ise Formation |

Outcrops of the Afowo Formation in some localities consist of oil sands in areas of dense vegetation where road cuttings expose them. They can also be found along river banks. The basal part of Afowo Formation is comprised of mixed brackish to marine facies alternating with well-sorted, sub-rounded sandy facies, suggesting a littoral or estuarine depositional environment [24,25]. The lower part of the formation was assigned to a Turonian age based on its palynological assemblage, with the upper part extending into the Maastrichtian. Afowo Formation has also been given a Cenomanian-Coniacian age [21] based on the planktonic foraminiferal species of Rotalipora Greenhornensis.

The Araromi Formation overlies the Afowo Formation, which means it is the youngest Cretaceous sediment in the basin. The lithological units of the formation consist of fine-to-medium-grained sandstone at the basal part, which is overlain by shales, siltstones, and interbeds of limestone, marl, and lignite [7].

Eight samples consisting of four oil sands and four plant roots were collected and used for the study. The oil sands were acquired from the two different boreholes, *vide infra*, and a sample each of *Theobroma cacao*(cocoa), *Cola* (kola nut), *Elaeis guineensis* (oil palm), and *Citrus* (orange) roots were collected from the immediate vicinities of the boreholes. The oil sands were acquired from the borehole's sidewall with a hand digger, while the plant samples were collected with hand trowels. All tools were washed, cleaned, and dried with a paper towel to avoid cross-contamination.

The citrus root was obtained close to Ilubinrin's borehole location, whereas the cocoa, kola, and oil palm roots were collected around the Ode-Aye borehole. All the samples were collected, wrapped in aluminum foil, and then placed in airtight sample bags, which were labeled to aid sample identification. The samples were then transported to the laboratories in sealed bags and preserved at room temperatures for further analysis.

The oil sands were analyzed for their hydrocarbon composition using the thermal desorption gas chromatography (TD-GC) Agilent 7890A system at the Agat Laboratories, Calgary, Canada. In addition, they were treated and assessed for chromium (Cr), nickel (Ni), vanadium (V), lead (Pb), cobalt (Co), and cadmium (Cd) with a Unicam Solar M-series Atomic Absorption Spectrophotometer at the Biodiversity and Conservation Biology Laboratory, University of the Western Cape, South Africa.

The plants were treated and analyzed for the same metals with a Unicam Solar M-series Atomic Absorption Spectrophotometer. The analytical procedure for determining the heavy metals in the oil sands and plant roots included acid digestion.

The general requirements for testing and calibration standard EPA method 3050 [26] and the compendium of methods for determination of toxic organic compounds EPA/625/R-96/010b [27] were used for the analytical measurements.

*2.1. Thermal Desorption-Gas Chromatography (TD-GC)*

The hydrocarbon composition of the oil sands was determined with an Agilent 7890A Gas Chromatography. The sample composition was assessed by separating the mixtures

into their components. A mass of 100 mg of oil sand was introduced into the instrument's inlet and heated to a temperature of 340 °C. The hydrocarbons were vaporized and captured by the cryogenic trap, where they were rapidly heated and then introduced to the capillary column. The individual hydrocarbon components were separated in the column based on their physical properties, such as boiling point with the lightest molecular fraction eluting first and the heaviest fraction eluting last. The flame ionization detector (FID) identified the separated components and quantified the relative proportion of each fraction.

## 2.2. Acid Digestion

The plant roots were dried in an oven at 60 °C for at least 72 h until stable weights were attained. Subsequently, the dried roots were milled, homogenized, and prepared for acid digestion. Acid digestion was carried out on the oil sands and the milled plant roots using an Aqua Regia digestion method [28]. Three blank samples were also prepared for quality assurance. The eleven digestion tubes with samples and blanks containing 9 mL of each acid mixture were digested in an aluminum alloy heating block at 110 °C for 4 h in a fume cupboard. An additional 9 mL of the acid mixture was added to ensure complete digestion after a further 3 h. Finally, the tubes were removed from the heating block and allowed to cool to room temperature inside the fume cupboard. The digested samples were diluted, filtered, and the volume was made up to 100 ml with deionized water.

## 2.3. Atomic Absorption Spectrometry

The heavy metal contents of the digested samples were analyzed with a Unicam Solar M-series Atomic Absorption Spectrophotometer. The aqueous solution was introduced into the instrument using a capillary tube and then atomized by the graphite furnace. The atomized elements were subjected to optical radiation from which absorption was identified and quantified to provide the concentrations of the elements using EPA method 3050.

## 3. Results

The thermal desorption-gas chromatography (TD-GC) results for the oil sands and plants are presented in Table 2 and Figures 3–6. The hydrocarbon fractions from the TD-GC were categorized as light condensates (nC6 to nC12), heavy condensates (nC13 to nC40), naphthenes (i.e., cyclopentane, methyl cyclopentane, cyclohexane, and ethyl cyclohexane), and aromatics (i.e., benzenes, toluene, ethylbenzene, and xylenes). The total composition of light condensate in the oil sand ranged from 0.40 to 2.03%, while the heavy condensate varied from 95.32 to 98.32%. The naphthenes and aromatics ranged from 0.00 to 0.40%.

**Table 2.** TD-GC hydrocarbon composition summary.

| Sample ID | Formation | Sample Type | Sample Depth (m) | Paraffins | | % Naphthenes | % Aromatics |
|---|---|---|---|---|---|---|---|
| | | | | % Light Condensate | % Heavy Condensate | | |
| OD-1 | Afowo | Oil Sand | 2.50 | 2.03 | 95.32 | 0.11 | 0.08 |
| OD-2 | Afowo | Oil Sand | 4.10 | 0.58 | 96.59 | 0.00 | 0.00 |
| IL-1 | Afowo | Oil Sand | 2.00 | 0.40 | 98.32 | 0.00 | 0.00 |
| IL-2 | Afowo | Oil Sand | 3.10 | 0.59 | 97.54 | 0.40 | 0.07 |

OD = Ode Aye-1 well; IL = Ilubinrin-1 well.

The A.A.S. results of the heavy metals shown in Table 3 and Figures 7 and 8 illustrated varying concentrations of heavy metals in the oil sands; Cr (1.686–5.733 ppm, with a mean value of 2.957 ppm), Ni (4.005–11.716 ppm, with a mean of 7.006 ppm), V (2.699–7.708 ppm, with a mean of 4.557 ppm), Pb (0.649–0.978 ppm, with a mean of 0.845 ppm), Co (0.953–3.223 ppm, with a mean of 1.770 ppm), and Cd (0.188– 0.461 ppm, with a mean of 0.277 ppm). Conversely, these metals were in varying concentrations in the *Citrus*, *Theobroma Cacao*, *Elaeis guineensis*, and Cola; Cr (7.095–32.685 ppm, with a

mean of 14.235 ppm), Ni (4.981–32.432 ppm, with a mean of 15.814 ppm), V (0.551–11.983 ppm, with a mean of 3.866 ppm), Pb (0.001–0.190 ppm, with a mean of 0.062 ppm), Co (0.695–4.425 ppm, with a mean of 2.457 ppm), and Cd (0.023–0.262 ppm, with a mean of 0.179 ppm) for the plant roots.

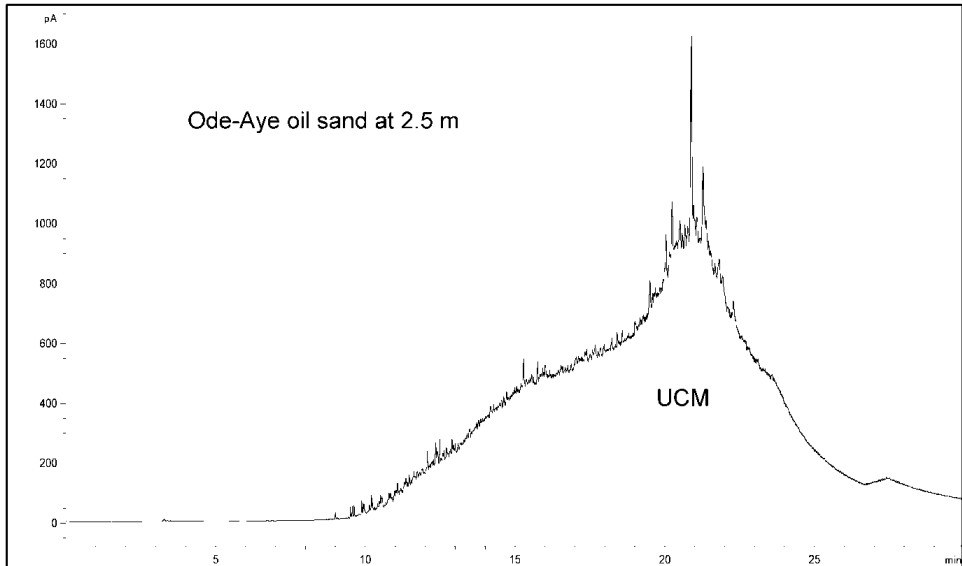

**Figure 3.** Chromatogram of Ode-Aye-1 oil sand at a depth of 2.5 m. The response intensity (pA) is displayed on the *y*-axis, while the retention time is shown on the *x*-axis in minutes. The chromatogram showed the hump of the unresolved complex mixture (UCM) below the peaks, suggesting biodegraded oil.

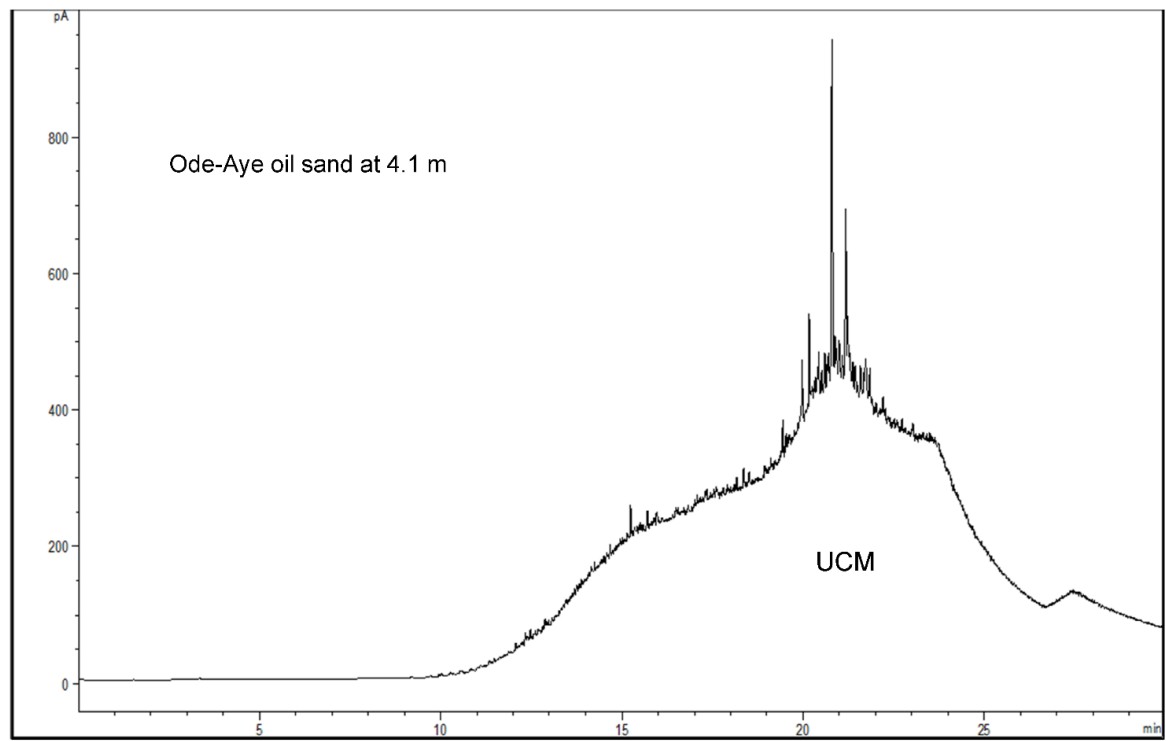

**Figure 4.** TD-GC result showing the chromatogram of Ode-Aye-1 oil sand at a depth of 4.1 m. The response intensity (pA) is displayed on the *y*-axis, while the retention time is shown on the *x*-axis in minutes. The chromatogram illustrated an unresolved complex mixture (UCM) below the peaks, indicating biodegraded oil.

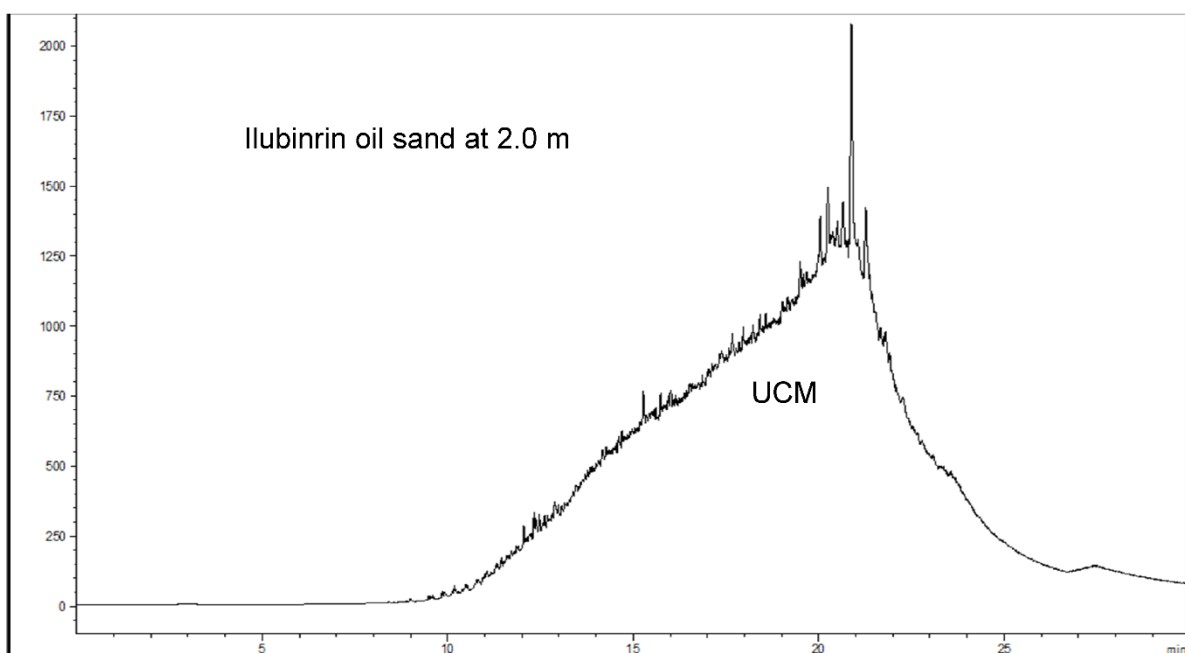

**Figure 5.** TD-GC result showing the chromatogram of Ilubinrin-1 oil sand at a depth of 2.0 m. The response intensity (pA) is displayed on the *y*-axis, while the retention time is shown on the *x*-axis in minutes. The chromatogram showed an unresolved complex mixture (UCM) below the peaks, indicating petroleum biodegradation.

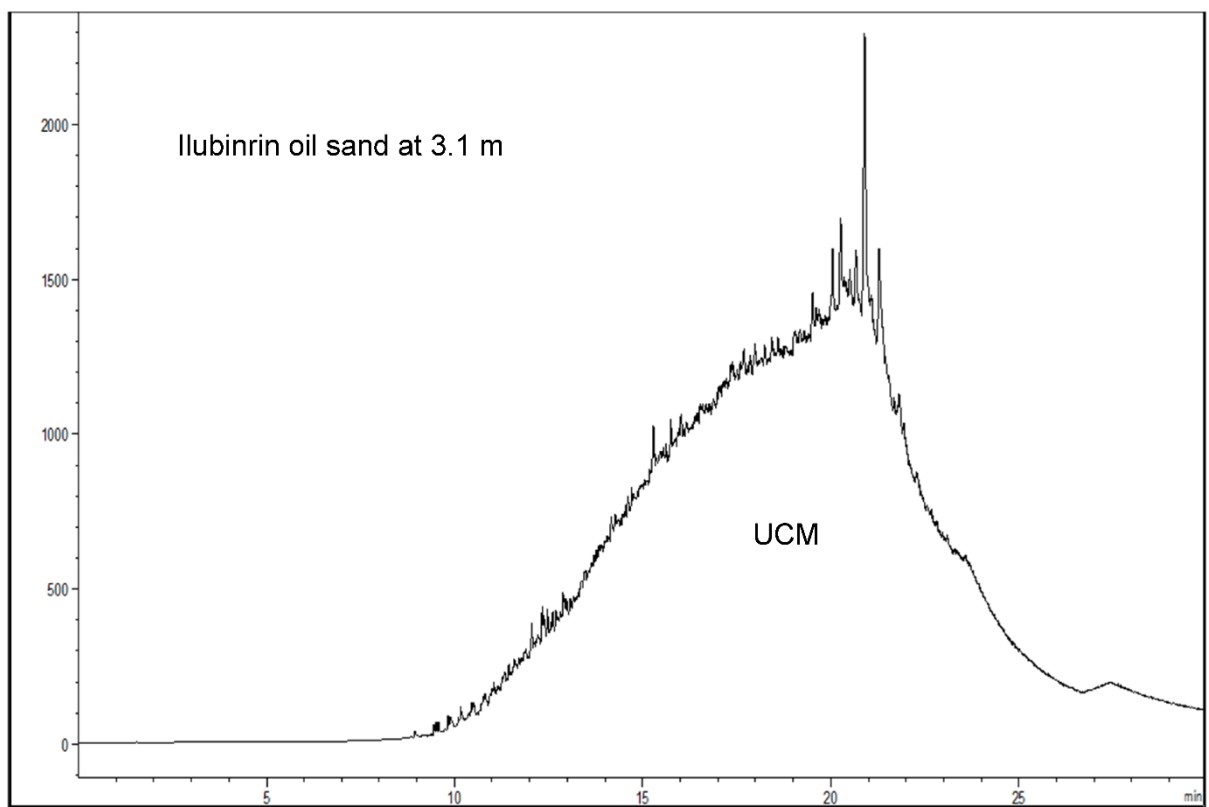

**Figure 6.** TD-GC result showing the chromatogram of Ilubinrin-1 oil sand at a depth of 3.1 m. The response intensity (pA) is displayed on the *y*-axis, while the retention time is shown on the *x*-axis in minutes. The presence of unresolved complex mixture (UCM) below the peaks suggests biodegraded oil.

**Table 3.** The concentration of chromium (Cr), nickel (Ni), vanadium (V), lead (Pb), cobalt (Co), and cadmium (Cd) in oil sands and plant roots.

| Sample ID Oil Sands | Sample Depth (m) | Cr (ppm) | Ni (ppm) | V (ppm) | Pb (ppm) | Co (ppm) | Cd (ppm) |
|---|---|---|---|---|---|---|---|
| OD-1 | 2.5 | 5.733 | 11.716 | 7.708 | 0.940 | 3.223 | 0.191 |
| OD-2 | 4.1 | 2.201 | 6.537 | 2.699 | 0.978 | 1.651 | 0.271 |
| IL-1 | 2.0 | 1.686 | 4.005 | 3.377 | 0.649 | 0.953 | 0.188 |
| IL-2 | 3.1 | 2.208 | 5.767 | 4.445 | 0.815 | 1.273 | 0.461 |
| **Range** | | 1.686–5.733 | 4.005–11.716 | 2.699–7.708 | 0.649–0.978 | 0.953–3.223 | 0.188–0.461 |
| Mean±SD | | 2.957 ± 1.866 | 7.006 ± 3.313 | 4.557 ± 2.220 | 0.845 ± 0.148 | 1.77 ± 1.006 | 0.277 ± 0.128 |
| **Plant Roots** | | | | | | | |
| Orange | N/A | 7.475 | 4.981 | 0.551 | 0.001 | 0.806 | 0.177 |
| Cocoa | N/A | 7.095 | 16.697 | 2.151 | 0.023 | 3.942 | 0.254 |
| Oil Palm | N/A | 32.685 | 32.423 | 11.983 | 0.190 | 4.425 | 0.262 |
| Kola | N/A | 9.687 | 9.157 | 0.779 | 0.037 | 0.695 | 0.023 |
| Range | | 7.095–32.685 | 4.981–32.423 | 0.551–11.983 | 0.001–0.190 | 0.695–4.425 | 0.023–0.262 |
| Mean±SD | | 14.235 ± 12.352 | 15.814 ± 12.087 | 3.866 ± 5.457 | 0.062 ± 0.086 | 2.457 ± 1.507 | 0.179 ± 0.122 |

Note: SD = Standard deviation. OD-1 = oil sand at a depth of 2.5 m from Ode-Aye-1 well. OD-2 = oil sand at a depth of 4.1 m from Ode-Aye-1 well. IL-1 = oil sand at a depth of 2.0 m from Ilubinrin-1 well. IL-2 = oil sand at a depth of 3.1 m from Ilubinrin-1 well.

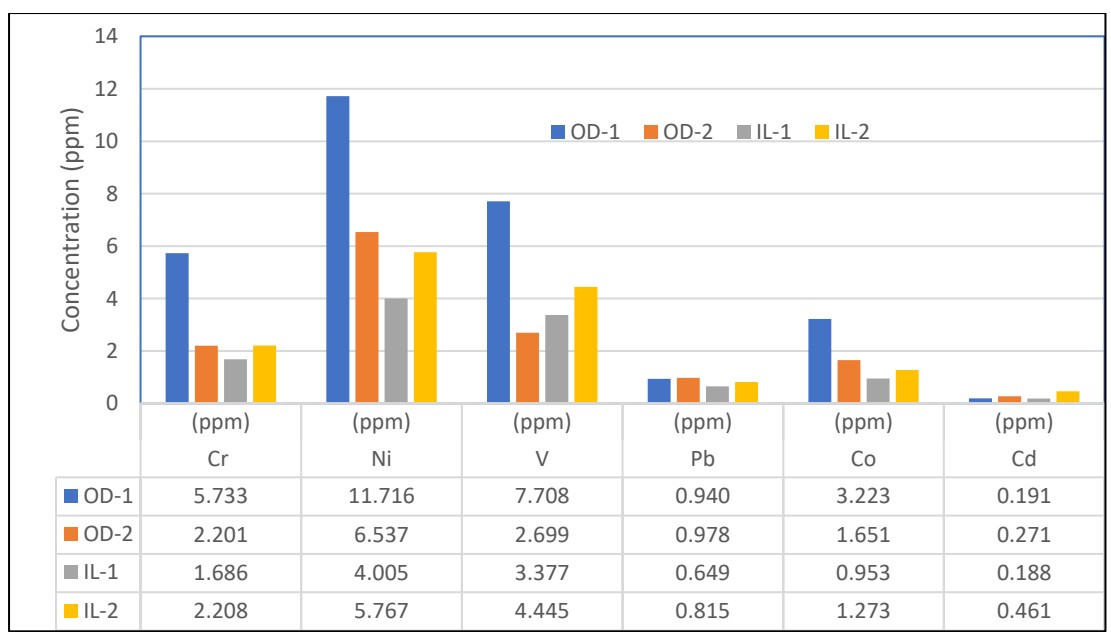

**Figure 7.** The concentration of Cr, Ni, V, Pb, Co, and Cd in oil sands.

The concentrations of Cr, Ni, V, Pb, Co, and Cd in the *Citrus* were 7.475, 4.981, 0.551, 0.001, 0.806, and 0.177 ppm, respectively. *Theobroma Cacao* showed Cr, Ni, V, Pb, Co, and Cd concentrations of 7.095, 16.697, 2.151, 0.023, 3.942, and 0.254 ppm, respectively. *Elaeis guineensis* also showed the presence of Cr (32.685 ppm), Ni (32.423 ppm), V (11.983 ppm), Pb (0.190 ppm), Co (4.425 ppm), and Cd (0.262 ppm). The amounts of Cr, Ni, V, Pb, Co, and Cd in the *Cola* were 9.687, 9.157, 0.779, 0.037, 0.695, and 0.023 ppm, respectively.

The results of the heavy metal analysis indicated that the uptake and bioaccumulation of Cr, Ni, V, Pb, Co, and Cd in the plant's roots are nonanthropogenic but related to the oil sand deposits in the area. All the metals in the oil sands were also identified in the plants.

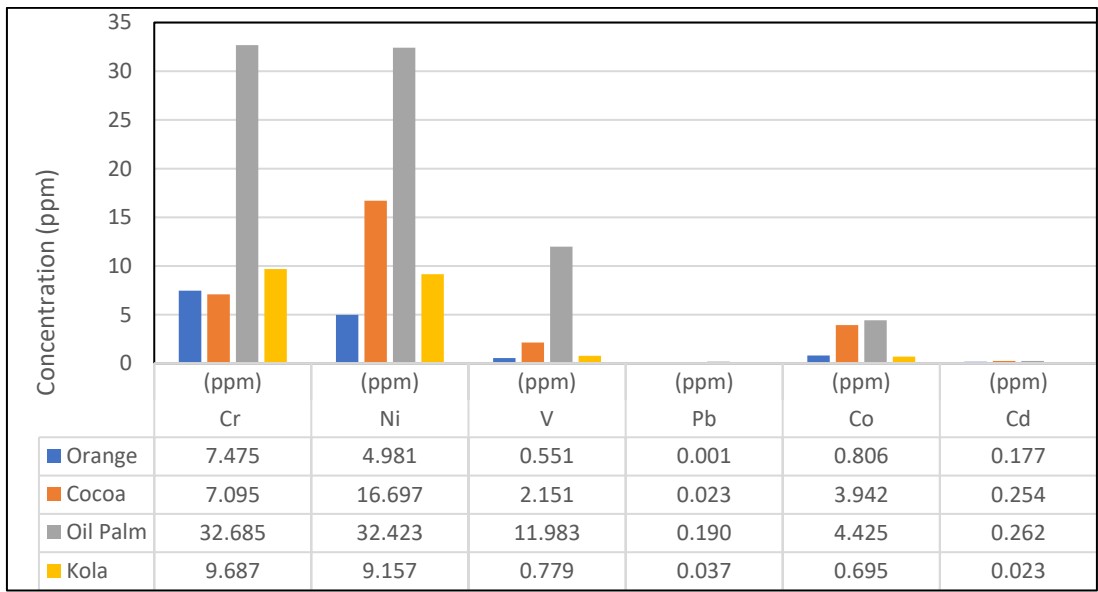

| | (ppm) Cr | (ppm) Ni | (ppm) V | (ppm) Pb | (ppm) Co | (ppm) Cd |
|---|---|---|---|---|---|---|
| Orange | 7.475 | 4.981 | 0.551 | 0.001 | 0.806 | 0.177 |
| Cocoa | 7.095 | 16.697 | 2.151 | 0.023 | 3.942 | 0.254 |
| Oil Palm | 32.685 | 32.423 | 11.983 | 0.190 | 4.425 | 0.262 |
| Kola | 9.687 | 9.157 | 0.779 | 0.037 | 0.695 | 0.023 |

**Figure 8.** The concentration of Cr, Ni, V, Pb, Co, and Cd in the taproots of orange, cocoa, oil palm, and kola trees.

## 4. Discussion

The results of this study provide critical evaluation parameters to assess the hydrocarbon composition, extent of petroleum alteration, and heavy metal contents of the oil sands and associated economic plants.

Hydrocarbons owe its genesis to the pyrolysis of kerogen in the source rocks. The ultimate composition of the kerogen is influenced by the source materials (organic and inorganic) in the sediments. Some of these materials include metals, which are released and transported during the generation and migration of the hydrocarbons from the source rock. The thermal evolution of Type I and II kerogen yields light crude oil with an API gravity of 30° to 40° [10]. Once the oil has accumulated in the reservoir, its quality can be drastically changed, and this degradation is attained through several processes and, in some instances, it can be so severe as to alter the characteristics of the crude oil forever. Degradation of the reservoired hydrocarbons results in the accumulation of the heavy end molecular components with low API gravity, high viscosity, high sulfur, and high metal concentrations [11]. The variation in the concentration of metals in crude oil indicates a genetic relationship between unaltered and completely altered crude oils [29].

The hydrocarbon composition analysis of the oil sand assessed in this study (Table 2, Figures 3–6) showed that it is mainly composed of heavy fractions (95.32 to 98.32%). By contrast, the light condensates varied from 0.40 to 2.03 0%. In addition, the oil sand chromatograms illustrated the humps of the unresolved complex mixture (UCM) below the peaks, and their response intensity showed no detectable hydrocarbon below 10 min retention time. The composition and the chromatogram's features suggested that the oil sands evaluated in this study are severely biodegraded which may have been accelerated by their shallow depths of occurrence, low temperature and salinity, and water washing. The heavy seasonal rainfalls in southwestern Nigeria from July to November may have contributed to the severity of the degradation through water washing.

The ultimate composition of petroleum may be strongly influenced by alteration after accumulation [30], resulting in the enrichment of heavy metals in biodegraded crude oil [11]. The oil sands in this study exhibited characteristics of extensive biodegradation, and the A.A.S. results showed the presence of V, Ni, Cr, Co, Pb, and Cd (Table 3 and Figures 7 and 8). The World Health Organization's (WHO) recommended safe limits for Cd, Cr, Ni, and Pb in agricultural soil are 0.003, 0.1, 0.05, and 0.1 ppm, respectively [31–34]. The permissible limits for these metals were exceeded by the oil sands (Table 2) from the two locations.

The concentrations of Cr in the oil sands varied from 1.686 to 5.733 ppm. The amounts of Ni, Pb, and Cd ranged from 4.005–11.716, 0.649–0.978, and 0.188–0.461 ppm, respectively. The severity of the hydrocarbon alteration illustrated by TD-GC is also supported by the geochemical data from the A.A.S.

Plants can absorb heavy metals from soils through their roots and leaves with root uptake being the most significant pathway for bioaccumulation [35,36]. All the plant's roots in this study showed wide-ranging concentrations of V, Ni, Cr, Co, Pb, and Cd identified in the oil sand in their vicinity, reflecting bioaccumulation. The sequence of accumulation in the *Elaeis guineensis* followed the descending order of Cr > Ni > V > Co > Cd > Pb. In contrast, *Theobroma cacao*, the bioaccumulation sequence was Ni > Cr > Co > V > Cd > Pb. The accumulation sequences for *Citrus* and *Cola* were in the descending order of Cr > Ni > Co > V > Cd > Pb and Cr > Ni > V > Co > Pb > Cd, respectively. The concentrations of Cr (32.685 ppm), Ni (32.423 ppm), V (11.983 ppm), Co (4.425 ppm), Cd (0.262 ppm), and Pb (0.190 ppm) were highest for *Elaeis guineensis*, suggesting an increased susceptibility for bioaccumulation.

Heavy metals, such as lead, cadmium, nickel, mercury, and chromium, are potentially hazardous either in combined or elemental forms [31]. They are toxic to plants, animals, and humans in high concentrations [37–41]. In addition, metals are not biodegradable, which means their bioaccumulation in living organisms causes biological and physiological complications [42]. The permissible limits for Cd, Cr, Pb, and Ni in plants of the World Health Organization, Food and Agricultural Organization, and the Department of Petroleum Resources Nigeria are 0.02, 1.30, 2.00, and 10.00 ppm, respectively [43]. Except for *Cola*, with a concentration of Cd (0.023 ppm) and Ni (9.687 ppm), all the other plants in this study exceeded the thresholds for Cd and Ni. The amounts of Cr in *Citrus* (7.475 ppm), *Theobroma cacao* (7.095 ppm), *Elaeis guineensis* (32.685 ppm), and *Cola* (9.687 ppm) exceeded the safe Cr threshold.

Of all the heavy metals, cadmium is considered to have the highest toxicity to plants and is carcinogenic to humans. It accelerates oxidative damage, disrupts metabolism, damages morphology and physiology, and reduces uptake and transportation of nutrients and water to plants [44]. In addition, toxic derivates of chromium damage chlorophyll [45], thus impeding the plant's vitality. Plant growth, metabolism, and sugar transport are also affected by nickel toxicity [46]. The toxicity of Cd, Cr, Ni, and other heavy metals identified in this study may be responsible for these plants' mortality, thereby confirming the farmers' observation that their crops are "not thriving and are dying".

The metal–metal correlation of the plant's roots to oil sand indicates that their origins are not due to human activities [47]. Therefore, the source of these metals is related to the oil sand deposits in the Dahomey Basin.

## 5. Conclusions

The results of this study indicate that the oil sands are severely biodegraded and comprised mainly of heavy-end molecular components, and they showed characteristic features of profoundly altered crude oil (i.e., humps of the unresolved complex mixture in the chromatograms, missing C6-C12 hydrocarbon chains, and absence of pristane and phytane). In addition, all the oil sands and plants illustrated the presence of chromium, nickel, vanadium, lead, cobalt, and cadmium.

The observations from this study can be summarized as follows:

(1) The TD-GC results indicated severely altered oil. The hydrocarbon composition analysis showed a dominance of heavy-end hydrocarbon fractions, and the chromatograms showed no detectable hydrocarbon below 10 min retention time due to the absence of light-end fractions. In addition, the chromatograms illustrated the classical humps of the unresolved complex mixtures (UCM), indicating heavily altered crude oil.

(2) The geochemical data from the A.A.S showed a close relationship in the concentrations of Cr, Ni, V, Pb, Co, and Cd in the oil sands and plants, suggesting a good correlation.

(3)　The concentration of the metals in the oil sand agreed with the biodegraded signatures presented by gas chromatography, implying that the metal concentration in crude oil may increase with increasing biodegradation.

(4)　The World Health Organization's (WHO) recommended safe and permissible limits for these metals in agricultural soils were exceeded by the oil sands.

(5)　The toxic effects of the heavy metals absorbed by the plants may have accelerated their mortality.

Further studies are recommended to determine the concentration and toxicity of heavy metals in the stems, leaves, fruits, and seeds of *Theobroma Cacao, Cola, Coffea arabica, Elaeis guineensis*, *Citrus*, and similar farm produce from the surrounding areas. In addition, any new study should include samples from locations devoid of oil sands for comparative analysis.

**Author Contributions:** Conceptualization, S.M.; methodology, S.M., M.O. and S.T.; software, S.M., S.T. and L.C.; validation, S.M., M.O. and S.T.; formal analysis, S.M., S.T. and M.O. and L.C.; writing— original draft preparation, S.M.; writing—review and editing, S.M., M.O. and S.T.; supervision, M.O. and S.T. All authors have read and agreed to the published version of the manuscript.

**Funding:** This research received no external funding.

**Data Availability Statement:** Not applicable.

**Acknowledgments:** The authors are grateful to the Ministry of Mines and Steel Development, Nigeria, and the host communities of Ilubinrin and Ode Aye, who ensured that the field works went unencumbered. In addition, we are grateful to the late Chief Rufus Olapade of Ilubinrin and Chief Akindoye of Ode Aye for their help and generosity. We are also thankful to Bamigbose James and Musa Aruna. Finally, Agat Laboratories, Calgary, Canada, is gratefully acknowledged for the work on TD-GC.

**Conflicts of Interest:** The authors declare no conflict of interest.

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
