# Peer review of "Metal–Metal Correlation of Biodegraded Crude Oil and Associated Economic Crops from the Eastern Dahomey Basin, Nigeria"

_minerals, doi:10.3390/min12080960_

Round 1

Reviewer 1 Report

Manuscript entitled “Metal-Metal Correlation of Biodegraded Crude Oil and Associ- 2 ated Economic Crops from the Eastern Dahomey Basin, Nigeriasubmitted by Saeed Mohammed, Mimonitu Opuwari, Salam Titinchi and Lilburne Cyster, can be considered for publication in Minerals Journal, after a serious major revision.

Here is a list of my specific comments:

1.     General comment 1: The utility of this study should be clearly highlighted in the manuscript.

2.     General comment 2: Pay attention on the interpretation of the experimental results, in accordance with the main objectives of this study. Only their presentation is not enough for a scientific paper.

3.     General comment 3: Figures should be rearranged in the manuscript.

4.     Page 1, lines 12-16: “The Eastern Dahomey Basin in southwestern…”. These abstract should be deleted.

5.     Page 1, Keywords: The number of keywords should be reduced.

6.     Page 2, line 49: “While the oil…”. Add here a reference.

7.     Page 2, line 63: “…the vicinity of the boreholes (Figure 1).”. Move this figure in the next section, as well as the paragraph “The first borehole (Ode Aye-1) was sited…”.

8.     Page 2, line 91: “The study's overarching aim…”. At the end of Introduction, the main objectives of this study should be clearly and detailed presented.

9.     Page 2, 2. Geological Setting: This section should be reorganized. Pay attention on technical details and provide a clear description of the geology of this zone. Delete general observations/comments, because are irrelevant here.

10.  Figures 6-8 should be moved into Supplementary materials.

11.  Page 8, 3. Materials and Methods: (a) The section “2. Geological Setting” should be included here. (b) The experimental methodology used in this study should be clearly presented.

12.  Page 14, 5. Discussion: All experimental results presented in the previous section should be detailed discussed here.

13.  Page 15, 6. Conclusions: Include in this section the most important experimental results and findings to highlight the importance of this study.

14. Page 16, References: The number of references is too high and should be reduced. 

Author Response

Dear Reviewer,

Many thanks for your constructive comments. I have attached the response to your comments.

Kind regards,

Opuwari

Reviewer 2 Report

Dear authors,

Thank you very much for inviting me to review the manuscript " Metal-Metal Correlation of Biodegraded Crude Oil and Associated Economic Crops from the Eastern Dahomey Basin, Nigeria”. The manuscript is very interesting, but from my humble point of view, it requires some changes and explanations that should be made previously that the manuscript will be published in Minerals Journal

1.- Introduction:

- Literature review, in my point of view is weak, which required to improve and strengthen. On the other hand, authors need to cite more latest researches in the relevant field to provide an up-to-date picture of work.

- Would you please add a few sentences in the Introduction section about that heavy metal pollution constitutes one of today’s most significant environmental problems (you can refer to: https://doi.org/10.1016/j.jclepro.2020.121608).

- The main objective of the paper must be written on the more clear way at the end of this section.

2.- Geological Setting:

- Figure 3 must be improved since the text of it cannot be readable.

2.- Materials and Methods:

- What standards were followed to analyzed the oil sands? All standards and regulatory specifications used in the present study should appear in the references.

3.- Results and discussion:

-  Indicate direction for future research.

- Figure 13 must be improved since the legend of it cannot be readable.

4.- Conclusions:

- Discuss the relevance of your work.

5.-Others:

- It would be convenient for the manuscript to be reviewed by a native English speaker so that any grammatical errors it may contain are corrected.

Author Response

Dear Reviewer,

We received your constructive comments on our manuscript with thanks. I have attached our responses to your comments.

Kind regards,

Opuwari

Round 2

Reviewer 1 Report

Manuscript entitledMetal-Metal Correlation of Biodegraded Crude Oil and Associ- 2 ated Economic Crops from the Eastern Dahomey Basin, Nigeriasubmitted by Saeed Mohammed, Mimonitu Opuwari, Salam Titinchi and Lilburne Cyster, can be considered for publication in Minerals Journal, after a major revision.

Here is a list of my specific comments:

  1. Page 1, 1. Introduction: This section should be detailed. The most important aspects related to this topic should be presented in order to provide a clear descrition of the state of art in this field.
  2. Page 3, 2. Geological Setting: This section should be included into Experimental section. Also, pay attention on technical details and provide a clear description of the geology of this zone. Delete general observations/comments, because are irrelevant here.
  3. Page 7, Figures 4 and 5: These figures should be moved into Supplementary materials.
  4. Page 14, 5. Discussion: This section should be detailed. All experimental results presented in the previous section should be clearly and detailed discussed in this section.
  5.  Page 15, 6. Conclusions: Include in this section the most important experimental results and findings to highlight the importance of this study.

Author Response

Dear Editor,

May I firstly express my sincere appreciation for the review of our manuscript. The authors are grateful to anonymous reviewers for their constructive comments. We took the comments seriously, and our point-by-point responses are below, using red colour text to denote reviewer 1 comments and black colour as our response.

Reviewer #1:

  1. Page 1, 1. Introduction: This section should be detailed. The most important aspects related to this topic should be presented in order to provide a clear descrition of the state of art in this field.

Suggestion implemented. See revised manuscript, lines 50-52, 57-69, and 80-84.

  1. Page 3, 2. Geological Setting: This section should be included into Experimental section. Also, pay attention on technical details and provide a clear description of the geology of this zone. Delete general observations/comments, because are irrelevant here.

                     Suggestion implemented

  1. Page 7, Figures 4 and 5: These figures should be moved into Supplementary materials.                  

Suggestion implemented. See the revised clean manuscript. Figures 4 and 5 were removed from the manuscript

  1. Page 14, 5. Discussion: This section should be detailed. All experimental results presented in the previous section should be clearly and detailed discussed in this section.

        Suggestions implemented. See the abstract of the revised clean manuscript.

  1. Page 15, 6. Conclusions: Include in this section the most important experimental results and findings to highlight the importance of this study

        Suggestion implemented. See the revised manuscript.

Thank you very much.

Reviewer 2 Report

This reviewer commends the authors efforts in addressing the comments. The responses are satisfactory, and the manuscript is hereby recommended for publication.

Author Response

Thank you very much for your positive feedback. This is much appreciated.